

# A Comparison of 3-D Spherical Shell Thermal Convection results at Low to Moderate Rayleigh Number using ASPECT (version 2.2.0) and CitcomS (version 3.3.1)

Grant T. Euen[1], Shangxin Liu[1,2], Rene Gassmöller[2], Timo Heister[3], and Scott D. King[1]

[1]Department of Geosciences, 926 West Campus Drive, Virginia Tech, Blacksburg, VA, USA
[2]Department of Geological Sciences, 241 Williamson Hall, University of Florida, Gainesville, FL, USA
[3]Mathematical and Statistical Sciences, O-110 Martin Hall, Clemson University, Clemson, SC, USA

**Correspondence:** Grant T. Euen (egrant93@vt.edu) and Shangxin Liu (sxliu@vt.edu)

**Abstract.** Due to the increasing availability of high-performance computing over the past decades numerical models have become an important tool for research in geodynamics. Several generations of mantle convection software have been developed, but due to their differing methods and increasing complexity it is important to evaluate the accuracy of each new model generation to ensure published geodynamic research is reliable and reproducible. We here explore the accuracy of the open-source,

finite-element codes ASPECT and CitcomS as a function of mesh spacing using low to moderate Rayleigh number models in steady-state, thermal convection. ASPECT (Advanced Solver for Problems in Earth's ConvecTion) is a new generation mantle convection code that enables modeling global mantle convection with realistic parameters and complicated physical processes using adaptive mesh refinement (Kronbichler et al., 2012; Heister et al., 2017). We compare the ASPECT results with calculations from the finite element code CitcomS (Zhong et al., 2000; Tan et al., 2006; Zhong et al., 2008), which has a long history

of use in the geodynamics community. We find that the globally-averaged quantities: RMS velocity, mean temperature, and Nusselt number at the top and bottom of the shell, agree to within 1%, and often much better, for calculations with sufficient mesh resolution. We also show that there is excellent agreement of the time-evolution of both the RMS velocity and the Nusselt numbers between the two codes for otherwise identical parameters. Based on our results we are optimistic that similar agreement would be achieved for calculations performed at convective vigor expected for Earth, Venus, and Mars.

## 1 Introduction

While there have been significant efforts to develop software capable of modeling mantle convection in a 3-D spherical shell (e.g. Baumgardner, 1985; Bunge et al., 1996; Ratcliff et al., 1996; Zhong et al., 2000; Kageyama and Sato, 2004; Yoshida and Kageyama, 2004; Choblet, 2005; Stemmer et al., 2006; Tan et al., 2006; Choblet et al., 2007; Tackley, 2008; Stadler

et al., 2010; Burstedde et al., 2013; Davies et al., 2013), there are few detailed comparison studies of results from more than





one code. The modeling software CitcomS has a long history of use in mantle convection studies (e.g., Zhong et al., 2000; Tan et al., 2002; McNamara and Zhong, 2004; Roberts and Zhong, 2004; McNamara and Zhong, 2005; Zhong, 2006; Tan et al., 2006; King, 2008; Foley and Becker, 2009; Sekhar and King, 2014; Liu and Zhong, 2015; King, 2018) and has been compared with analytic kernel solutions and other published results using thermal convection at low Rayleigh number (Zhong

et al., 2008). ASPECT is a new generation, massively-parallel, mantle convection code combining Adaptive Mesh Refinement (AMR) technology with modern numerical methods (Kronbichler et al., 2012; Heister et al., 2017), built on top of the deal.II finite element library (Bangerth et al., 2007; Arndt et al., 2021). Three distinct features set ASPECT apart from most other mantle convection codes: (1) its governing equations are dimensional and are written to allow both incompressible and fully compressible flow to be calculated; (2) AMR technology combined with linear and nonlinear solvers allows users to perform

mesh adaptation with various refinement or coarsening strategies; (3) second-order finite elements are employed to discretize the velocity and temperature in the domain, which should lead to better accuracy for a given number of degrees of freedom and a better convergence rate with increasing resolution (c.f. Kronbichler et al., 2012; Heister et al., 2017).

There have been a number of studies comparing ASPECT results with other codes using Cartesian geometry (Kronbichler et al., 2012; Tosi et al., 2015; Puckett et al., 2017; Heister et al., 2017; He et al., 2017; Glerum et al., 2018), however, ASPECT

has not yet had a systematic benchmark using a 3-D spherical shell geometry. Both the solvers for incompressible Boussinesq Stokes flow and thermal convection of CitcomS have been systematically benchmarked in 3-D spherical shell geometry. The ASPECT solver for incompressible Boussinesq Stokes flow has been benchmarked through analytical propagator matrix solutions (Liu and King, 2019) and a new family of special analytical solutions at spherical harmonic degree 1 and order 0 (Thieulot, 2017). However, the accuracy of the thermal convection calculations (i.e., the energy equation) of ASPECT in 3-D

spherical shell geometry has not been tested. No resolution studies of thermal convection in a 3-D spherical-shell have been reported for either CitcomS or ASPECT.

In this work we report a comparison of steady-state thermal convection at low to moderate Rayleigh number using both CitcomS and ASPECT. A number of previous studies have focused on the low Rayleigh number calculations ($7 \times 10^3$) with viscosity variations up to a factor of $10^5$ (Zhong et al., 2008, and references therein). Zhong et al. (2008) also includes cal-

culations of Rayleigh number $10^5$, a more moderate value, with viscosity variations up to a factor of 30. These allow for steady-state solutions that facilitate comparison between codes. In this work we include both Rayleigh number $7 \times 10^3$ and $10^5$ calculations. We report the Rayleigh number using the traditional definition where the length-scale ($D^3$) is the thickness of the spherical shell rather than the radius of the planet. In addition, we reproduce these calculations on a number different resolution meshes to document the convergence of the globally-averaged diagnostics of the steady-state temperature and velocity fields

including Root-Mean-Square (RMS) velocity, mean temperature, and Nusselt number at the inner and outer boundaries of the shell.



## 2 Method

The conservation of mass, momentum, and energy equations for an incompressible, Boussinesq fluid in their non-dimensional forms are given by

$$0 = \nabla \cdot \boldsymbol{v}, \tag{1}$$

$$0 = -\nabla P + \nabla \left[ \eta (\nabla \boldsymbol{v} + \nabla \boldsymbol{v}^T) \right] + \mathrm{Ra}\hat{g}, \tag{2}$$

$$0 = \frac{\partial T}{\partial t} + \boldsymbol{v} \cdot \nabla T - \nabla^2 T, \tag{3}$$

where $t$ is time, $\boldsymbol{v}$ is velocity, $P$ is pressure, $\hat{g}$ is the radial unit vector pointing toward the center of the planet, and $T$ is temperature (Schubert et al., 2001).

The Rayleigh number alone can describe this problem if all material properties and gravity are held constant. The Rayleigh
number is given by,

$$\mathrm{Ra} = \frac{\rho \alpha g \Delta T D^3}{\kappa \eta}, \tag{4}$$

where $\rho$ is the density, $\alpha$ is the coefficient of thermal expansion, $g$ is gravity, $\Delta T$ is the change in temperature across the domain, $D$ is the depth of the domain, $\kappa$ is the thermal diffusivity, and $\eta$ is the dynamic viscosity. Because ASPECT by default solves the equations in dimensional form, but this benchmark is calculated using a non-dimensional scaling, we report the
15 parameters used in the ASPECT calculations to achieve a Rayleigh number of $7 \times 10^3$ and $10^5$ in Table 1. Boundary conditions are set to be free-slip for the inner and outer shell velocity. Temperature is set to a constant $T = 0$ on the outer boundary and $T = 1$ on the inner boundary. The thickness of the shell is set to 0.45, with an inner boundary radius, $r_b$, of 0.55 and an outer boundary radius, $r_t$, of 1.0. By using these values, as well as values of 1 for most other parameters, the Rayleigh number can be controlled by the value of gravity alone (Table 1).

For the ASPECT calculations, we use version 2.2.0 (Bangerth et al., 2020b; Kronbichler et al., 2012; Heister et al., 2017; Bangerth et al., 2020a) published under the GPL2 license to solve equations (1-4) using the Boussinesq formulation option. For CitcomS we use version 3.3.1 (Tan et al., 2006; Zhong et al., 2000; McNamara and Zhong, 2004; Moresi et al., 2014) also published under the GPL2 license. Both codes are available from the github repository of the Computational Infrastructure for Geodynamics (CIG).

The cases that we consider use temperature-dependent, non-dimensional viscosity expressed as

$$\eta = e^{E(0.5 - T)}, \tag{5}$$

where $E$ is a viscosity parameter similar to activation energy and $T$ is temperature. Following the model naming convention used in Zhong et al. (2008), the letter $A$ refers to cases with Rayleigh number $7 \times 10^3$ in a tetragonal steady-state and the letter $C$ refers to cases with Rayleigh number $10^5$ in a cubic steady-state. The numbers following the letter represent each individual
case, which differ by their total variation in viscosity, $\Delta\eta$. We focus on a limited number of viscosity variations, ranging from $\Delta\eta = 1$, or constant viscosity, to $\Delta\eta = 10^5$. The value of $\Delta\eta$ for each case tested in this study is reported in Table 2.





We compare the results from ASPECT and CitcomS on a variety of meshes and we report the top and bottom Nusselt number, mean temperature, and RMS velocity. The Nusselt number, Nu, is the ratio of convective to conductive heat transfer normal to the boundary of the domain. We report the top and bottom Nusselt numbers, defined as

$$\text{Nu}_t = \frac{r_t(r_t - r_b)}{r_b}Q_t, \tag{6}$$

and

$$\text{Nu}_b = \frac{r_b(r_t - r_b)}{r_t}Q_b, \tag{7}$$

where $Q_t$ and $Q_b$ are the surface and bottom heat fluxes, $r_b = 0.55$, and $r_t = 1.0$. We also report the mean temperature and the spherically-averaged RMS velocity. The volume of the spherical domain is given by

$$\Omega = \frac{4\pi(r_t^3 - r_b^3)}{3}. \tag{8}$$

This makes the mean temperature

$$\langle T \rangle = \frac{1}{\Omega}\int_\Omega T d\Omega, \tag{9}$$

and the spherically-averaged RMS velocity

$$\langle V_{\text{RMS}} \rangle = \left[\frac{1}{\Omega}\int_\Omega v^2 d\Omega\right]^{\frac{1}{2}}. \tag{10}$$

The values of the RMS velocity, mean temperature, and top and bottom Nusselt numbers are averaged over the same non-

dimensional time intervals as those reported in Zhong et al. (2008) (Table 2).

ASPECT uses quadratic velocity and temperature elements by default and has the capability to refine the mesh based on a variety of measured properties of the solution. CitcomS by comparison, uses linear velocity and temperature elements with a mesh spacing that remains fixed throughout the calculation. The authors of Zhong et al. (2008) refined the CitcomS mesh at the outer and inner boundaries of the shell. In contrast, we use a uniformly-spaced mesh in the radial direction. We test meshes

at various refinements, including higher levels than those used by Zhong et al. (2008). In order to have a more systematic view of how increasing resolution improves model accuracy, we chose not to refine the CitcomS mesh in our calculations.

To facilitate the comparison between ASPECT and CitcomS we performed the calculations in this work with a mesh having constant cell spacing in the radial direction. This allowed isolate differences between the two codes stemming from their different numerical methods, as opposed to different mesh structures. AMR and other mesh refining/coarsening strategies for

ASPECT are turned off unless otherwise stated.

In order to more accurately reproduce the CitcomS results, the rheology (Eq. 5) was added to ASPECT as a stand-alone plugin, which is possible because ASPECT is written to allow adding new features without modifying the main source code itself. By writing and compiling plugins, a user can modify existing features, or add completely new ones. A complete description of how to write plugins is available in the ASPECT manual (Bangerth et al., 2020b). The source code for the specific





plugins used here can be found at VTechData (https://data.lib.vt.edu - DOI to be minted with paper is finalized) . Specifically, one plugin was written to implement a Frank-Kamenetskii rheology as a standalone material model, and a second plugin was written to allow multiple spherical harmonic perturbations to be used simultaneously as initial conditions. This was necessary to reproduce the cubic-planform cases later in the study.

For CitcomS we used the default parameter setting in the CitcomS-3.3.1 version from CIG with the following exceptions: down_heavy and up_heavy, which are the number of smoothing cycles for down-ward/upward smoothing, are set to 3; vlow-step and vhighstep, which are the number of smoothing passes at lowest/highest levels are set to 30 and 3 respectively; and max_mg_cycles is the maximum number of multigrid cycles per solve and is set to 50. Our experience showed that fewer down-ward/upward smoothing cycles lead to time-dependent results for some meshes while all the other meshes achieved

steady solutions. For example, the $12 \times 32 \times 32 \times 32$ mesh for C1 was time-dependent with down_heavy and up_heavy set to 2 whereas when down_heavy and up_heavy are set to 3, the solution was steady as it was for all other meshes. Increasing these parameters had no discernible impact on the overall run time. We caution the reader that the calculation did not converge using the default setting of these parameters in the 3.3.1 version, therefore recommend users set down_heavy and up_heavy to 3.

CitcomS requires the user to specify the coarsest mesh and number of multigrid levels with the formula for each direction

being

$$nodex = 1 + nprocx \times mgunitx \times 2^{levels-1} \tag{11}$$

where nprocx is the number of processors in the x dimension, mgunitx is the size of the coarsest mesh in the multigrid solver, and levels is the number of multigrid levels. For each mesh we use at least three multigrid levels, as experience shows that fewer multigrid levels can lead to convergence problems. The parameters that we use for each mesh is shown in Table 3. Using

different parameters leads to small differences in the final global quantities reported in Tables 5-10.

## 3   Results

The default ASPECT temperature solver is the entropy viscosity (EV) method (Guermond et al., 2011; Kronbichler et al., 2012). The ASPECT team implemented a Streamline-Upwind Petrov Galerkin (SUPG) advection-diffusion solver (Brooks and Hughes, 1982) as a part of this work. The SUPG algorithm is also implemented in CitcomS (Zhong et al., 2000) and Con-

Man (King et al., 1990). ASPECT has several benchmarks included to test robustness of these advection stabilization methods. One test of an advection-diffusion solver is to advect a pattern of known shapes in a 2-D box and rotate them 360 degrees at a prescribed velocity (see Figure 130 in Bangerth et al. (2020b)). Another test was created using a simple, 4-cell convection pattern in an annulus (Figure 1). These tests show that the solution using EV is surprisingly diffusive when using more coarse meshes. SUPG on the other hand shows much less diffusion even when coarse meshes are used. With sufficient mesh refine-

ment, the solutions from the two advection stabilization methods are almost identical. This is essentially a non-issue when performing tests in 2-D, as mesh resolution can be adjusted higher without significant change in computational difficulty or run time. However, for 3-D spherical tests, increasing resolution can cause a significant increase in the required computational





resources, making highly refined models infeasible. This means that the EV solution is more diffusive for the refinements typically used in 3-D spherical calculations. The ASPECT results shown here primarily use the SUPG implementation, which was part of the ASPECT 2.2.0 release. In the ASPECT 2.2.0 release, the EV parameters have been updated and the results are significantly improved over the results from older versions. In the tables we report the EV results for selected cases.

## 3.1 Low-Rayleigh-number, Tetragonal-Planform, Steady-State Thermal Convection

In this section we focus on the Rayleigh-number $7 \times 10^3$, tetragonal-planform, steady-state, thermal convection cases labeled A1-A9 in Zhong et al. (2008). The A cases use the same Rayleigh number and initial condition; the label 1-9 refers to the viscosity contrast. Results for the first three cases, A1-A3, were also reported in Ratcliff et al. (1996), Yoshida and Kageyama (2004), and Stemmer et al. (2006).

To create a tetragonal pattern, a degree 3, order 2 spherical harmonic perturbation is used. The magnitude of this perturbation for both the cosine and sine terms is $\epsilon_c = \epsilon_s = 0.01$. The final steady-state pattern of the temperature isotherms can be seen in Figure 2a and b. The four plumes represent the four corners of a uniform tetrahedron, hence we refer to this as a tetragonal-planform.

To assess how each code handles temperature-dependent rheology, we selected three cases: A1 (constant viscosity), A3 ($\Delta \eta = 20$), and A7 ($\Delta \eta = 10^5$). The constant viscosity case provides a baseline result without the added complexity of temperature-dependent rheology. Case A3 (Figure 2b) was chosen because its viscosity is weakly temperature-dependent and can be compared with published results from a number of mantle convection codes. Case A7 was chosen because with this large viscosity contrast the flow transitions into a stagnant-lid mode of convection, causing a much more complex planform (Figure 2c). Each case was run with both codes using multiple mesh refinement levels. The results of these runs were then used to extrapolate the theoretical results of a "mesh of infinite refinement" using a Richardson extrapolation. We computed each case using both the default spherical-shell ASPECT mesh and the radially-uniform mesh. For CitcomS we used a uniform vertical mesh spacing, which differs slightly from the refined mesh spacing at the top and bottom boundaries used in Zhong et al. (2008). We confirm that we can reproduce the output flow diagnostics reported in Zhong et al. (2008) when using the CitcomS-3.3.1 version downloaded from CIG with the exact parameters used in Zhong et al. (2008). Results for these three cases can be found in Tables 5-7.

The results from A1 and A3 on the CitcomS and uniform radial spacing ASPECT meshes are well-resolved and in good agreement. Case A7 has larger differences between the two codes, but the overall results are still well-resolved and steady. Plots of radially-averaged (averaged over shells of constant radii) horizontal and vertical velocity and temperature also show excellent agreement between both codes (Figure 3).

### 3.1.1 3-D Results for the Constant and Weakly Temperature-Dependent Viscosity Cases: A1 and A3

We compare the convergence of the solutions from CitcomS and ASPECT for the A1 cases (Figure 2a) by comparing the RMS velocity, mean temperature, and top and bottom Nusselt numbers on a series of increasingly refined meshes. For our ASPECT calculations we considered the radially-uniform and mesh. We use global mesh resolutions of 8, 16, 32, and 64 radial cells to





test convergence of model values. For our CitcomS calculations we use 16, 24, 32, 48, 64, and 96 radial elements. Throughout the document we use cells to describe the ASPECT meshes and elements to describe the CitcomS meshes because this is how the grids are described in the documentation. For CitcomS, where the mesh is divided into 12 cubic regions, each cube has the same number of elements on each side, thus the 16-radial element mesh is comprised of $12 \times 16 \times 16 \times 16$ elements. For

comparison, the results reported in Zhong et al. (2008) were calculated on a $12 \times 32 \times 32 \times 32$ mesh with increased refinement at the outer and inner shell boundaries. To facilitate the comparison between our present work and Zhong et al. (2008), we reproduce the the results reported in Zhong et al. (2008) in Table 5.

The plots of RMS velocity, mean temperature, and Nusselt numbers at the outer and inner shell boundaries on different meshes for each code (Figure 4) share a number of common features. For each code, as we increase the mesh size the values of

mean temperature, RMS velocity, top and bottoms Nusselt number converge. The top and bottom Nusselt numbers converge to within 0.07% for CitcomS and 0.01% for ASPECT, which is to be expected as the top and bottom Nusselt numbers should be equal if the codes conserve energy. The ASPECT mesh produces nearly identical results, with differences only appearing well-past the number of significant figures reported by Zhong et al. (2008). Radially-averaged values also show nearly identical solutions (Figure 3). We then extrapolate the values to an infinitesimal mesh using a Richardson extrapolation (Table 5).

These extrapolations are slightly different than the values determined by Zhong et al. (2008); however, this is not surprising because Zhong et al. (2008) reported the values from a single $32^3$ mesh with refinement near the surface and the base with no extrapolation to an infinitesimal mesh spacing. The ASPECT results for coarse meshes are closer to the extrapolated value than the CitcomS results for the same mesh, which is not surprising because ASPECT uses second-order elements while CitcomS uses first-order elements. One might argue that the 32-radial-cell ASPECT results should be compared with the 64 element

CitcomS results. We note that for ASPECT cases A1, the Entropy viscosity results are almost identical to, and in some cases superior to, the SUPG results (Table 5).

We also show that in addition to the small differences in the steady-state global quantities between the two codes, the time-series evolution of the global diagnostics follow nearly identical paths. Figure 5 shows RMS velocity against both Nusselt numbers for two ASPECT calculations and one CitcomS calculation of Case A1. The path taken to arrive at the solution is the

same for all calculations. The specific refinement of the mesh and the code used determines the exact values calculated, but the behavior of the solutions between both codes is consistent.

For the A3 cases (Figure 2b) we compare the results using the same meshes and diagnostics described for A1 (Table 6). Case A3 is very similar to A1, but has a weakly temperature-dependent viscosity, $\Delta\eta = 20$ (Table 2). Still, the tetragonal-planform is maintained throughout the run. As before, higher mesh refinements allow for greater convergence in both codes. The top

and bottom Nusselt numbers converge within 0.08% for CitcomS and within 0.01% for ASPECT. The ASPECT results follow a different evolution for convergence with increasing mesh resolution as those for CitcomS, except for the bottom Nusselt number (Figure 6). RMS velocity and bottom Nusselt number both taper to a high point. The pattern of the average temperature is slightly different, with the coarsest mesh refinement being slightly lower than the other data points. However, the top Nusselt number pattern includes a slightly high outlier at the 16 element (radial) mesh for ASPECT. CitcomS too has an outlier at the





48 element mesh, seen most prominently in RMS velocity. We note that for ASPECT cases A3, the Entropy viscosity results are almost identical to, and in some cases superior to, the SUPG results (Table 6).

### 3.1.2    3-D Results for the Stagnant-Lid Case: A7

Case A7 has the highest viscosity contrast of all of the cases reported here. Case A6 reported in Zhong et al. (2008) was

identified by the authors as a transitional state between mobile-lid and stagnant-lid behavior; all prior models had been mobile-lid, and all that followed were stagnant-lid. Case A7 was partly chosen because it falls into this category of stagnant-lid behavior. It also has the smallest $\Delta\eta$ of the three stagnant-lid cases, the other two being A8 and A9, so A7 was also chosen for its relative speed as the time to solve the velocity matrix depends on the viscosity contrast. Case A7 is calculated using ASPECT at mesh resolutions of 8, 16, and 32 radial elements.

Solutions for Case A7 have the most strongly-varying results based on the mesh refinement used of the A cases (Figure 7). This is to be expected, as this is the most computationally challenging run in this study. Though it is still amongst the low Rayleigh number cases, it has the largest viscosity contrast throughout the mantle.

Reported values for this case show more difference between the two codes, even at the highest mesh refinements tested (Table 7). Still, the two codes are in good agreement, with radially-averaged values from the most refined meshes of both

codes being nearly identical (Figure 3). More data points from further refined meshes would assist in establishing the pattern of convergence, however, at 32 radial elements the computation is already exceedingly costly. Case A7 run to completion using a refinement of 64 radial elements would take weeks of run time on 384 processors, hence these calculations were not performed. However, the difference between top and bottom Nusselt numbers still show that solutions are well-resolved. CitcomS converges to within 0.17% while ASPECT converges to within 0.11%. Isotherms from this case help to understand

the behavior that is not seen in the other cases. The initial, tetragonal pattern is lost in this case, as several, smaller plumes of hot material upwell throughout the mantle (Figure 2c). In Cases A1 and A3 the tetragonal pattern is maintained throughout the run (Figure 2a and b). These results match the behavior reported in Zhong et al. (2008). Looking at the steady global diagnostics reported, it can be seen that both codes are in good agreement, and ASPECT is very likely converging with higher mesh resolution for this parameter as well.

### 3.2    Intermediate-Rayleigh-number, Cubic-Planform, Steady-State Thermal Convection

The C cases have a similar setup to the A cases with a few notable differences. For their initial condition they use two spherical harmonic perturbations of degree 4, order 0 and degree 4, order 4 simultaneously, resulting in the number of plumes increasing from 4 to 6 (Figure 2d-f). If a cube is envisioned surrounding the model, these six plumes are evenly spaced at the centers of its sides, hence, the planform of the C cases is referred to as cubic. The Rayleigh number is also increased to $10^5$, compared

to the tetragonal-planform cases using a value of 7000. While this is still smaller than the Rayleigh number typically used in geodynamic models, it approaches the planetary range. Case C1 is a perfect analog to Case A1, using a constant viscosity ($\Delta\eta = 1$) with the new cubic-planform initial conditions. Cases C2 and C3 use weakly temperature-dependent viscosity, analogous to Case A3 ($\Delta\eta = 10$ and 30, respectively). However, they use the final state of C1 as their initial condition rather than starting





at time 0. For example, cases of C2 and C3 on an 8-radial-element mesh resolution used the final solution of Case C1 on an 8-radial-element resolution. ASPECT runs of C2 and C3 also kept the mesh type consistent for these initial conditions.

The results from all three C cases tested are well-resolved and in good agreement between the two codes. Case C2 proved slightly more challenging for CitcomS, but with higher mesh resolution good convergence was achieved (Figure 9). Radially-
averaged values also show strong agreement between both codes for all C cases (Figure 3).

### 3.2.1   3-D Results for the Constant Viscosity Case: C1

As with the tetragonal-planform cases, we compare the convergence of the solutions from CitcomS and ASPECT for the C1 cases (Figure 2d) by comparing the RMS velocity, mean temperature, and top and bottom Nusselt numbers on a series of increasingly refined meshes. For our ASPECT calculations we used the radially-uniform ASPECT mesh described previously.
We use global mesh resolutions of 8, 16, 32, and 64 radial cells to test convergence of model values. For our CitcomS calculations we use 24, 32, 48, 64, and 96 radial elements. Results reported in Zhong et al. (2008) were calculated on a $12 \times 48 \times 48 \times 48$ mesh with increased refinement at the outer and inner boundary of the spherical shell. To facilitate the comparison between our present work and Zhong et al. (2008), we reproduce the results reported in Zhong et al. (2008) in Table 8.

Plots of RMS velocity, mean temperature, and top and bottom Nusselt numbers on different meshes are again produced,
with overall convergence increasing with mesh refinement. The top and bottom Nusselt numbers converge to within 0.55% for CitcomS and 0.17% for ASPECT. It should be noted that our CitcomS run using 48-radial-element resolution has an agreement of 0.06%, an outlier in the convergence trend. Radially-averaged plots show strong agreement between the two codes (Figure 3). We then extrapolate the values to an infinitesimal mesh using a Richardson extrapolation (Table 8). Once again, the ASPECT results for coarse meshes are closer to the extrapolated value than the CitcomS results for the same mesh resolution. Agreement
between the two Nusselt numbers for each case actually follows a similar pattern of convergence between the two codes. Though by a resolution of 32-cell-radial ASPECT mesh is already within 0.3% agreement, while CitcomS is only within 2.5%.

Data points trend smoothly towards convergence for ASPECT, with a slight outlier at 8-radial-element resolution. The RMS velocity at that resolution especially falls farther from other data. We note that overall EV produces solutions with Nusselt numbers in better agreement for this case; however, its RMS velocity outlier at 8-radial-element resolution is larger. CitcomS
data points also trend towards convergence for both Nusselt numbers. RMS velocity and temperature have some outliers at coarse mesh resolutions, though at higher resolutions there is still a clear trend towards convergence. We note that for ASPECT cases C1, the Entropy viscosity results are almost identical to, and in some cases superior to, the SUPG results (Table 8).

### 3.2.2   3-D Results for the Weakly Temperature-Dependent Viscosity Cases: C2 and C3

Case C2 (Figure 2e) has a weakly temperature-dependent viscosity, $\Delta \eta = 10$ (Table 2). The top and bottom Nusselt numbers
converge within 0.6% for CitcomS and within 0.1% for ASPECT. The agreement between Nusselt numbers is markedly different for the two codes in this case (Table 9). ASPECT results at 8-cell-radial resolution show differences of 10%, significantly higher than any previous case. However, increased resolution caused a dramatic convergence, allowing for agreement as good as all previous cases reported. CitcomS results did not approach the difference observed with ASPECT on the most coarse





meshes, although the 24-radial-element and $32^3$-radial-element meshes produced values that both did not match the pattern of convergence of the more refined meshes.

The ASPECT results show well-behaved convergence for all parameters calculated (Figure 9). RMS velocity and temperature show very little change with increased resolution. Both Nusselt numbers show more change between resolutions, the top

Nusselt number changing the most, but both show very stable convergence. CitcomS shows more difficulty with this case. Results for all values show an initial increase through the coarse meshes which then begins to decrease and converge. All values show a similar change, with the bottom Nusselt number changing the least. The temperature of the CitcomS runs also returns to a similar level as produced by the coarse refined meshes. Despite this difficulty, radially-averaged plots show that the solutions of the most-refined meshes still agree to a high degree between CitcomS and ASPECT (Figure 3).

Case C3 (Figure 2f) has a slightly stronger temperature-dependent viscosity, $\Delta\eta = 30$. Top and bottom Nusselt numbers converge to within 0.1% for both CitcomS and ASPECT on both meshes. As in Case C2, ASPECT results for Case C3 at 8-radial-cell resolution have a surprisingly high difference between Nusselt numbers. But, once again, increased resolution strongly increases convergence. By the 32-radial-cell resolution, convergence is better resolved than Case C1 at constant viscosity. CitcomS seems to have less trouble with Case C3 than Case C2.

Convergence for both codes is good for all parameters tested. Nusselt numbers at the top and bottom especially show very high agreement between CitcomS and ASPECT, as well as the numbers reported by Zhong et al. (2008). Average temperature shows different time-series evolution between the two codes, but convergence is still achieved as resolution increases. RMS velocity shows the largest difference between the codes with increased resolution. Values from CitcomS and ASPECT are on different tracks, and while they do show overall convergence, it is not to the same degree as the other parameters. As with the

previous C cases though, radially-averaged plots show that the most well-resolved meshes produce highly similar solutions between CitcomS and ASPECT (Figure 3).

## 4   Conclusions

We note excellent agreement in the RMS velocity, mean temperature, and top and bottom Nusselt number between the two codes on the most refined meshes. If we take the difference between the top and bottom Nusselt numbers as a measure of

the accuracy of the solution, which should be zero for incompressible flow at steady state, the 16-radial-cell ASPECT mesh results are already within 1% for the A cases. CitcomS requires a 32-radial-element mesh to achieve the less than 1% difference between the top and bottom Nusselt numbers for Cases A1 and A3; Case A7 requires a 48-radial-element mesh.

For the higher Rayleigh number C cases, both codes need more refined meshes to achieve a 1% difference between the top and bottom Nusselt numbers. For ASPECT a 32-radial-cell mesh is needed while for CitcomS a 48-radial-cell mesh is needed.

We note that CitcomS shows some outlier cases where coarser meshes have unusually small percent differences between top and bottom Nusselt numbers. We use the overall pattern of convergence between the various mesh resolutions as a more accurate measure of the necessary refinement.

In general, globally-averaged diagnostics from both codes at the highest mesh resolutions tested agree to within 0.6% and the Richardson extrapolation of the results from increasing mesh resolution agrees to within 1.0%. Often the Richardson extrapolation agrees to within 0.5%. ASPECT generates better-resolved solutions on coarser meshes than CitcomS, as would be expected because it uses higher-order elements.

Many studies use CitcomS results computed on a 64-radial-element mesh, often with limited convergence checks on more refined meshes. For the intermediate C Family cases, we find that there is little difference between the 64-radial-element and 96-radial-element meshes, which generally supports the use of a 64-radial-element mesh.

Using the Streamline-Upwind Petrov Galerkin (SUPG) energy solver for ASPECT we find extremely good agreement between CitcomS and ASPECT results for these low to intermediate Rayleigh number calculations. Both CitcomS and ASPECT

use the SUPG algorithm to solve the energy equation. For the selected cases A1, A3, and C1 we find the Entropy Viscosity (EV) energy solver is as good and in many cases slightly better than the SUPG solver. We caution that these calculations have Nusselt numbers that are at least a factor of 3 smaller than anticipated values for Venus or Earth.

*Code availability.* All software used to generate these results are freely available. ASPECT is publicly available on GitHub at https://github.com/geodynamics/aspect, and can be found permanently at https://doi.org/10.5281/ZENODO.3924604. CitcomS is also publicly available

on GitHub at https://github.com/geodynamics/citcoms, and can be found permanently at https://doi.org/10.5281/zenodo.7271919.

*Author contributions.* Scott King, Grant Euen, and Shangxin Liu are responsible for initial conceptualization. Software updates for use of SUPG in ASPECT were developed by Timo Heister and Rene Gassmöller. ASPECT simulations were designed by Shangxin Liu. Models calculated using ASPECT were performed by Grant Euen and Shangxin Liu, and models calculated using CitcomS were performed by Scott King. Grant Euen performed the data curation and formal analysis, and prepared the manuscript with contributions from all co-authors.

*Competing interests.* The authors declare there are no competing interests in this work.

*Acknowledgements.* We thank the Computational Infrastructure for Geodynamics (http://geodynamics.org), which is funded by the National Science Foundation under awards EAR-0949446 and EAR-1550901, administered by University of California – Davis. Shangxin Liu was partially supported by EarthScope and GeoPRISMS programs of the National Science Foundation through the Mid-Atlantic Geophysical Integrative Collaboration (MAGIC) project of the grant EAR-1250988. Timo Heister was partially supported by the National Science

Foundation awards DMS-2028346, OAC-2015848, EAR-1925575, and by the Computational Infrastructure for Geodynamics, through the National Science Foundation under awards EAR-0949446 and EAR-1550901. Rene Gassmöller was partially supported by the National Science Foundation awards EAR-1925677, EAR-2054605, and by the Computational Infrastructure for Geodynamics, through the National Science Foundation under awards EAR-0949446 and EAR-1550901.





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



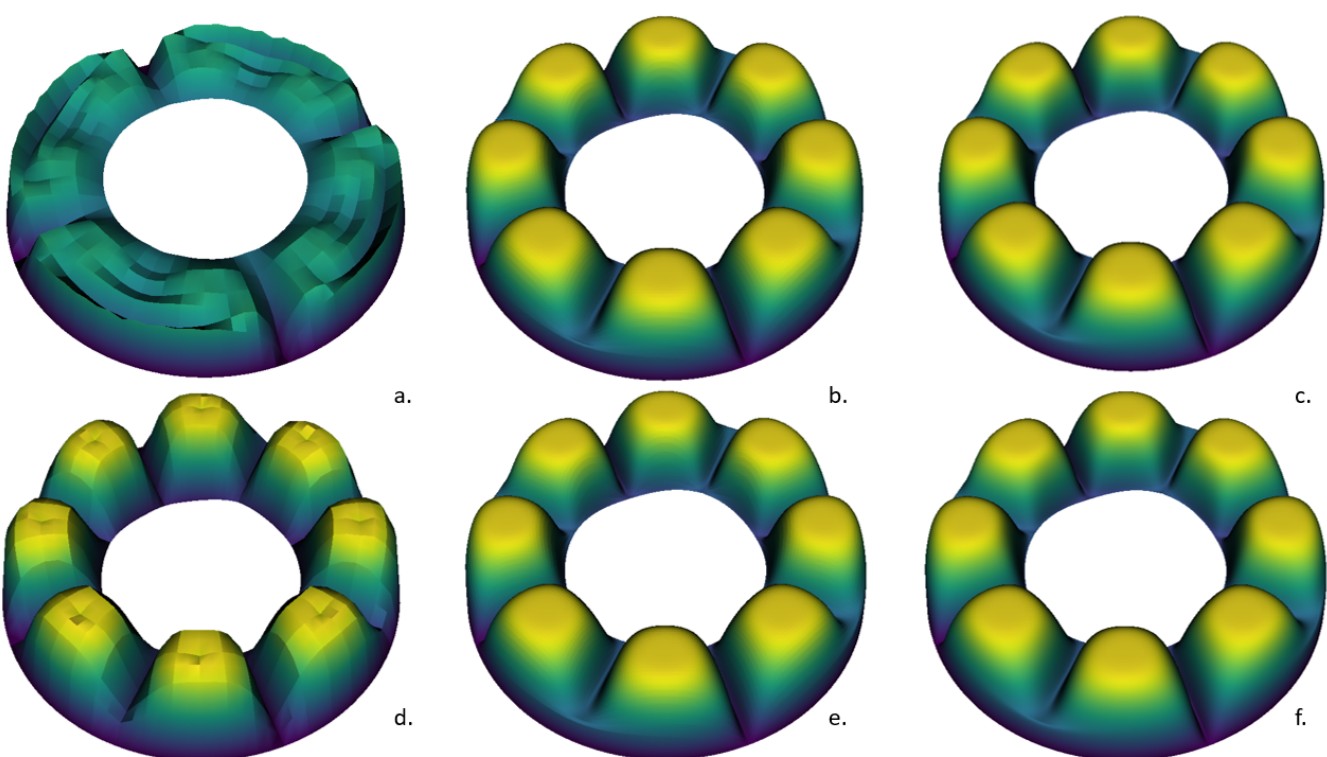

**Figure 1.** Results from the advection-in-annulus benchmark in ASPECT. This shows how mesh refinement influences the heat flux out of the system depending on whether Entropy Viscosity (a through c) or SUPG (d through f) is used. At moderate mesh refinement (b and e), and high mesh refinement (c and f), both solver schemes produce nearly identical results; however, coarser meshes (a and d) allow for very large differences in heat advection between the two methods. For models in 2-dimensions, this is not an issue, as very high refinement can be used without a major increase in computing cost. However, this is an issue for 3-dimensional models, as each increase in mesh refinement represents a significant increase in computational resources.



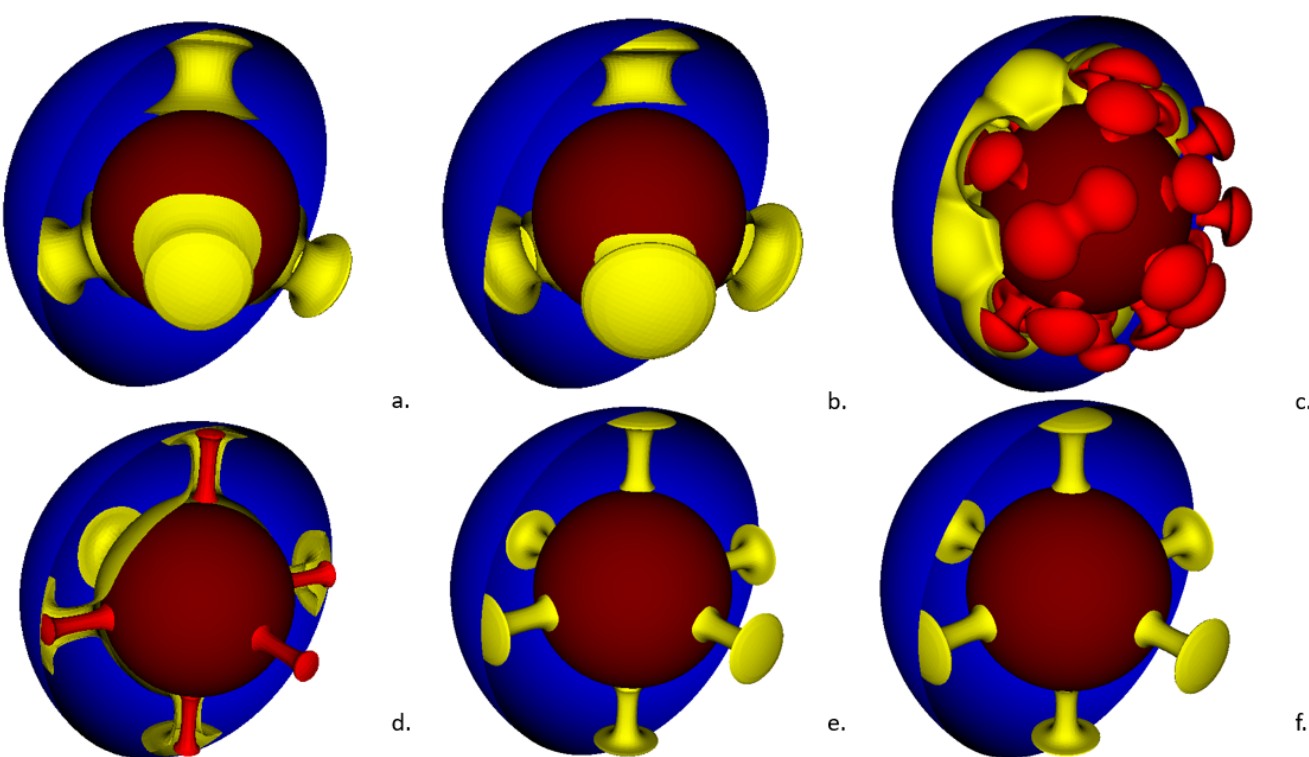

**Figure 2.** Isotherms from Cases A1, A3, A7, C1, C2, and C3 using a 32-cell radially-uniform mesh with ASPECT. Isotherms a, b and c are the tetragonal-planform cases A1, A3 and A7, respectively. Isotherms d, e and f are the cubic-planform cases C1, C2 and C3, respectively. For each image, the dark red, central sphere represents the core, the yellow plumes are hotter, upwelling material, and the blue half-shell is the surface. In A7 and C1 (c and d) there is also a brighter red layer, which is hotter material than the yellow layer. This is visualized to show more details for the complex convection of A7, and the cores of the plumes of C1. Isotherm values are 1 for dark red, 0.8 for bright red, 0.5 for yellow and 0.000001 (essentially 0) for blue.



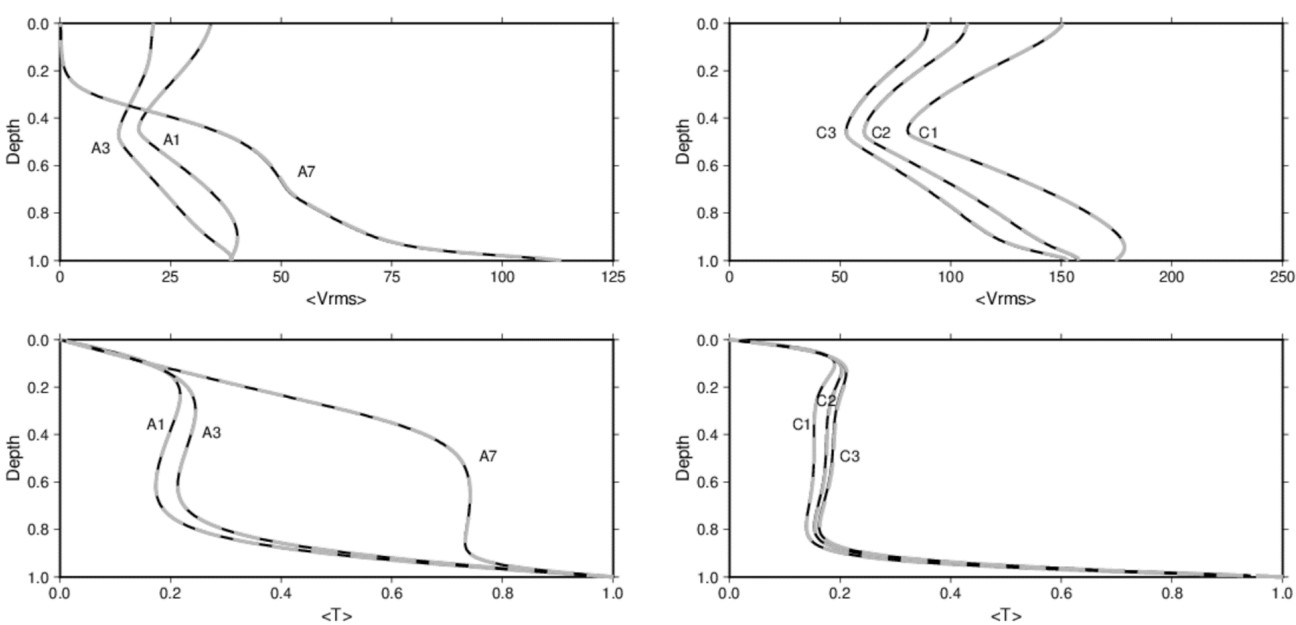

**Figure 3.** Plots of RMS velocity and average temperature with respect to depth for all cases tested. Dark gray, dashed lines are CitcomS data, black, solid lines are ASPECT data. All cases are in excellent agreement across both codes.





**Figure 4.** RMS velocity, average temperature and Nusselt number at the top and bottom of the model for Case A1 run at various refinements using CitcomS (red triangles), ASPECT using SUPG (blue stars), and ASPECT using EV (yellow dots). The values reported in Zhong et al. (2008) are also shown (black diamonds). Dashed lines are the extrapolated values for each code. ASPECT and CitcomS show strong agreement in their individual convergence paths with mesh refinement; only RMS velocity has a slight difference, though the difference is in the second decimal place.

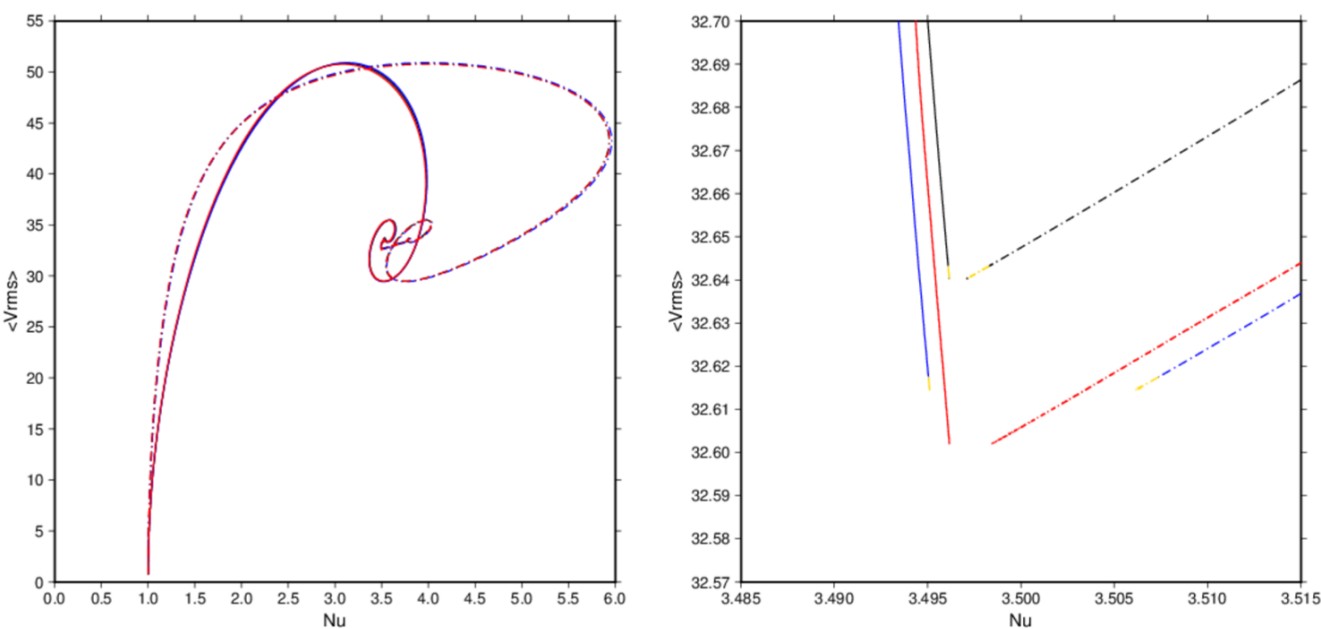

**Figure 5.** RMS velocity plotted against Nusselt numbers at both the top (dashed lines) and bottom (solid lines) of the model for Case A1 run using ASPECT at mesh refinements $12 \times 16^3$ (blue) and $12 \times 32^3$ (black), and CitcomS at mesh refinement $12 \times 96^3$ (red). In the ASPECT runs, gold segments represent the interval used in Zhong et al. (2008) to calculate averages (Table 2). These figures show the similarity in solution between the two codes. It also shows that the model is steady; most of the change happens in the early stage. As the solid and dashed lines approach each other they slow down. The averaging interval accounts for about $\frac{1}{3}$ of the model runs, but are only small pieces of each line.

**Figure 6.** RMS velocity, average temperature and Nusselt number at the top and bottom of the model for Case A3 run at various refinements using CitcomS (red triangles), ASPECT using SUPG (blue stars), and ASPECT using EV (yellow dots). The values reported in Zhong et al. (2008) are also shown (black diamonds). Dashed lines are the extrapolated values for each code. ASPECT and CitcomS show strong agreement in their individual convergence paths. RMS velocity shows a slight difference at the final convergence numbers, though it is small. CitcomS also has an outlier in the convergence path of RMS velocity at $12 \times 48^3$ (3rd point), which can be seen in the average temperature and top Nusselt number as well, though it is not as prominent.



**Figure 7.** RMS velocity, average temperature and Nusselt number at the top and bottom of the model for Case A7 run at various refinements using CitcomS (red triangles) and ASPECT (blue stars). The values reported in Zhong et al. (2008) are also shown (black diamonds). Dashed lines are the extrapolated values for each code. Of note are the ASPECT solutions for the bottom Nusselt number, which do not show as clear of a trend towards convergence as the other parameters. However, the values are in good agreement with CitcomS, and based on the other parameters very likely would converge with higher mesh resolution. Extrapolated values between the two codes show small differences for each parameter, larger than previous cases, though again, higher mesh resolution would likely cause these values to converge.



**Figure 8.** RMS velocity, average temperature and Nusselt number at the top and bottom of the model for Case C1 run at various refinements using CitcomS (red triangles), ASPECT using SUPG (blue stars), and ASPECT using EV (yellow dots). The values reported in Zhong et al. (2008) are also shown (black diamonds). Dashed lines are the extrapolated values for each code. ASPECT and CitcomS show strong agreement in all reported parameters. It should be noted that the coarsest meshes of ASPECT using EV (global refinement 2, yellow dots) are not shown for RMS velocity and average temperature. These values are higher than others (Table 8) and only make it more challenging to see the convergence of the rest of the data, so they were omitted from this figure.





**Figure 9.** RMS velocity, average temperature and Nusselt number at the top and bottom of the model for Case C2 run at various refinements using CitcomS (red triangles) and ASPECT (blue stars). The values reported in Zhong et al. (2008) are also shown (black diamonds). Dashed lines are the extrapolated values for each code. ASPECT and CitcomS show more of a difference in convergence in this case. CitcomS especially shows a very different path of convergence between its coarser and finer resolutions. Ultimately both codes come into better agreement at higher resolution, though parameters still show a larger difference than the other reported cases.

**Figure 10.** RMS velocity, average temperature and Nusselt number at the top and bottom of the model for Case C3 run at various refinements using CitcomS (red triangles) and ASPECT (blue stars). The values reported in Zhong et al. (2008) are also shown (black diamonds). Dashed lines are the extrapolated values for each code. ASPECT and CitcomS both show strong convergence in all parameters reported. Interestingly, both codes show better agreement than the previous case C2. Extrapolated values for RMS velocity and average temperature also show a larger difference from values reported in Zhong et al. (2008) than previous C cases.





**Table 1.** Parameters used in Zhong et al. (2008) experiments for ASPECT.

| Parameter | Symbol | Value |
|---|---|---|
| thickness of mantle | D | 0.45 |
| density | $\rho$ | $1.0\,\mathrm{kg/m^3}$ |
| temperature difference | $\Delta T$ | $1.0\,\mathrm{K}$ |
| thermal diffusivity | $\kappa$ | $1.0\,\mathrm{m^2\,s^{-1}}$ |
| thermal expansion coefficient | $\alpha$ | $1.0\,\mathrm{K^{-1}}$ |
| gravitational acceleration | g | $\frac{7000}{0.45^3}\,\mathrm{m\,s^{-1}}$ for the A cases |
| | | $\frac{100000}{0.45^3}\,\mathrm{m\,s^{-1}}$ for the C cases |
| reference viscosity | $\eta_0$ | $1.0\,\mathrm{Pa\,s}$ |
| reference temperature | $T_0$ | $0.5\,\mathrm{K}$ |
| velocity polynomial | | 2 |
| temperature polynomial | | 2 |
| Stokes tolerance | | $10^{-3}$ |
| CFL number | | 1.0 |





**Table 2.** Values of the A and C cases presented in this work taken from Zhong et al. (2008). The A cases used a 32-element radial mesh with refinement at the top and bottom. The C cases used a 48-element radial mesh with refinement at the top and bottom.

| Test Performed | Rayleigh Number | $\Delta\eta$ | $t_1$ | $t_2$ | $\langle V_{\mathrm{RMS}} \rangle$ | $\langle T \rangle$ | $\mathrm{Nu}_t$ | $\mathrm{Nu}_b$ |
|---|---|---|---|---|---|---|---|---|
| A1 | $7 \times 10^3$ | 1 | 0.7 | 1.0 | 32.66 | 0.2171 | 3.5126 | 3.4919 |
| A3 | $7 \times 10^3$ | 20 | 0.6 | 0.9 | 25.85 | 0.2432 | 3.1724 | 3.1548 |
| A7 | $7 \times 10^3$ | $10^5$ | 1.2 | 1.7 | 50.21 | 0.5039 | 2.7382 | 2.7431 |
| C1 | $1 \times 10^5$ | 1 | 0.255 | 0.315 | 154.8 | 0.1728 | 7.8495 | 7.7701 |
| C2 | $1 \times 10^5$ | 10 | 0.48 | 0.55 | 122.1 | 0.1908 | 7.0968 | 7.0505 |
| C3 | $1 \times 10^5$ | 30 | 0.52 | 0.57 | 109.1 | 0.2011 | 6.7572 | 6.7182 |



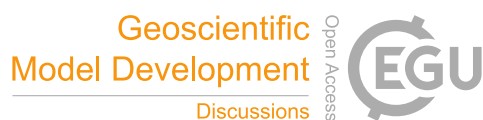

**Table 3.** The mesh structure used for CitcomS calculations. nodex, nodey, and nodez are the number of nodes in each direction for each of the 12 cubes making up the sphere. The formula for each direction is nodex = $1 +$ nprocx $\times$ mgunitx $\times 2^{levels-1}$ where nprocx is the number of processors in the x dimension, mgunitx is the size of the coarsest mesh in the multigrid solver, and levels is the number of multigrid levels.

| nodex,y,z | nprocx,y,z | mgunitx,y,z | levels |
|-----------|------------|-------------|--------|
| 17 | 1 | 4 | 3 |
| 25 | 1 | 3 | 4 |
| 33 | 1 | 8 | 3 |
| 49 | 1 | 6 | 4 |
| 65 | 2 | 8 | 3 |
| 97 | 2 | 6 | 4 |





**Table 4.** Degrees of Freedom (DoFs) for each resolution of each code. Note that Zhong et al. (2008) did not report the DoFs for their models. The values in parentheses under mesh resolution are the global refinement parameter. In ASPECT, this controls the starting mesh resolution.

| Code | Mesh Resolution | Velocity DoFs | Pressure DoFs | Temperature DoFs | Total DoFs |
|---|---|---|---|---|---|
| ASPECT | 8-radial-cells (global refinement 2) | 156,774 | 6,930 | 52,258 | 215,962 |
| | 16-radial-cells (global refinement 3) | 1,216,710 | 52,258 | 405,570 | 1,674,538 |
| | 32-radial-cells (global refinement 4) | 9,585,030 | 405,570 | 3,195,010 | 13,185,610 |
| | 64-radial-cells (global refinement 5) | 76,088,070 | 3,195,010 | 25,362,690 | 104,645,770 |
| CitcomS | 16-radial-elements ($12 \times 16^3$) | 142,848 | 47,616 | 47,616 | 238,080 |
| | 24-radial-elements ($12 \times 24^3$) | 487,296 | 162,432 | 162,432 | 812,160 |
| | 32-radial-elements ($12 \times 32^3$) | 1,161,216 | 387,072 | 387,072 | 1,935,360 |
| | 48-radial-elements ($12 \times 48^3$) | 3,939,840 | 1,313,280 | 1,313,280 | 6,566,400 |
| | 64-radial-elements ($12 \times 64^3$) | 9,363,456 | 3,121,152 | 3,121,152 | 15,605,760 |
| | 96-radial-elements ($12 \times 96^3$) | 31,684,608 | 10,561,536 | 10,561,536 | 52,807,680 |



**Table 5.** Results for Case A1 on all meshes tested. Column labeled '% diff' represents the percent difference between top and bottom Nusselt numbers for each case. A Richardson extrapolation was applied to the different data sets to estimate the values of a theoretical mesh of infinite refinement.

| Code | Case | $\langle V_{\mathrm{RMS}} \rangle$ | $\langle T \rangle$ | $\mathrm{Nu}_t$ | $\mathrm{Nu}_b$ | % diff |
|---|---|---|---|---|---|---|
| ASPECT | 8-radial-cells | 32.4304 | 0.215948 | 3.48413 | 3.48144 | 0.08 |
| | 16-radial-cells | 32.6152 | 0.215622 | 3.50645 | 3.49509 | 0.33 |
| | 32-radial-cells | 32.6403 | 0.215563 | 3.49710 | 3.49616 | 0.03 |
| | 64-radial-cells | 32.6431 | 0.215583 | 3.49651 | 3.49621 | 0.01 |
| | Extrapolated | 32.6437 | 0.215594 | 3.49653 | 3.49621 | 0.01 |
| ASPECT - EV | 8-radial-cells | 32.6431 | 0.215530 | 3.49097 | 3.49256 | -0.05 |
| | 16-radial-cells | 32.6431 | 0.215585 | 3.49653 | 3.49621 | 0.01 |
| | 32-radial-cells | 32.6431 | 0.215584 | 3.49653 | 3.49621 | 0.01 |
| | 64-radial-cells | 32.6431 | 0.215583 | 3.49650 | 3.49621 | 0.01 |
| | Extrapolated | 32.6431 | 0.215583 | 3.49649 | 3.49621 | 0.01 |
| CitcomS | 16-radial-elements | 32.4724 | 0.224335 | 3.50819 | 3.44182 | 1.93 |
| | 24-radial-elements | 32.6341 | 0.220444 | 3.53691 | 3.48379 | 1.52 |
| | 32-radial-elements | 32.6858 | 0.219212 | 3.53149 | 3.50174 | 0.85 |
| | 48-radial-elements | 32.7450 | 0.219826 | 3.53081 | 3.52230 | 0.24 |
| | 64-radial-elements | 32.6671 | 0.215989 | 3.50153 | 3.49592 | 0.16 |
| | 96-radial-elements | 32.6026 | 0.215615 | 3.49869 | 3.49616 | 0.07 |
| | Extrapolated | 32.5759 | 0.215568 | 3.49833 | 3.49711 | 0.03 |
| CitcomS | Zhong 2008 | 32.66 | 0.2171 | 3.5126 | 3.4919 | 0.59 |





**Table 6.** Results for Case A3 on all meshes tested. Column labeled '% diff' represents the percent difference between top and bottom Nusselt numbers for each case. A Richardson extrapolation was applied to the different data sets to estimate the values of a theoretical mesh of infinite refinement.

| Code | Case | $\langle V_{\mathrm{RMS}} \rangle$ | $\langle T \rangle$ | $\mathrm{Nu}_t$ | $\mathrm{Nu}_b$ | % diff |
|---|---|---|---|---|---|---|
| ASPECT | 8-radial-cells | 25.5987 | 0.241440 | 3.14525 | 3.13838 | 0.22 |
| | 16-radial-cells | 25.7395 | 0.241546 | 3.15895 | 3.15211 | 0.22 |
| | 32-radial-cells | 25.7623 | 0.241512 | 3.15362 | 3.15338 | 0.01 |
| | 64-radial-cells | 25.7661 | 0.241539 | 3.15386 | 3.15349 | 0.01 |
| | Extrapolated | 25.7672 | 0.241552 | 3.15413 | 3.15351 | 0.02 |
| ASPECT - EV | 8-radial-cells | 25.7423 | 0.241047 | 3.13389 | 3.14461 | -0.34 |
| | 16-radial-cells | 25.7593 | 0.241474 | 3.15272 | 3.15230 | 0.01 |
| | 32-radial-cells | 25.7655 | 0.241536 | 3.15378 | 3.15340 | 0.01 |
| | 64-radial-cells | 25.7661 | 0.241539 | 3.15386 | 3.15349 | 0.01 |
| | Extrapolated | 25.7662 | 0.241539 | 3.15387 | 3.15350 | 0.01 |
| CitcomS | 24-radial-elements | 25.7964 | 0.246309 | 3.19619 | 3.14644 | 1.58 |
| | 32-radial-elements | 25.7895 | 0.243007 | 3.17106 | 3.14899 | 0.70 |
| | 48-radial-elements | 25.6167 | 0.243622 | 3.15268 | 3.15197 | 0.02 |
| | 64-radial-elements | 25.7885 | 0.241954 | 3.15816 | 3.15318 | 0.16 |
| | 96-radial-elements | 25.7268 | 0.241558 | 3.15577 | 3.15336 | 0.08 |
| | Extrapolated | 25.6936 | 0.241432 | 3.15451 | 3.15340 | 0.04 |
| CitcomS | Zhong 2008 | 25.85 | 0.2432 | 3.1724 | 3.1548 | 0.56 |



**Table 7.** Results for Case A7 on all meshes tested. Column labeled '% diff' represents the percent difference between top and bottom Nusselt numbers for each case. A Richardson extrapolation was applied to the different data sets to estimate the values of a theoretical mesh of infinite refinement.

| Code | Case | $\langle V_{\mathrm{RMS}} \rangle$ | $\langle T \rangle$ | $\mathrm{Nu}_t$ | $\mathrm{Nu}_b$ | % diff |
|---|---|---|---|---|---|---|
| ASPECT | 8-radial-cells | 46.6868 | 0.487298 | 2.58091 | 2.73127 | -5.51 |
| | 16-radial-cells | 48.8020 | 0.497543 | 2.72461 | 2.72595 | -0.05 |
| | 32-radial-cells | 50.1224 | 0.510244 | 2.79869 | 2.80171 | -0.11 |
| | Extrapolated | 50.6329 | 0.515379 | 2.82677 | 2.83382 | -0.25 |
| CitcomS | 24-radial-elements | 52.2203 | 0.507014 | 2.74190 | 2.64551 | 3.64 |
| | 32-radial-elements | 51.0042 | 0.508720 | 2.76461 | 2.72934 | 1.29 |
| | 48-radial-elements | 50.3019 | 0.506870 | 2.78219 | 2.77310 | 0.33 |
| | 64-radial-elements | 50.3315 | 0.510222 | 2.79530 | 2.79053 | 0.17 |
| | Extrapolated | 50.3648 | 0.511768 | 2.80062 | 2.79703 | 0.13 |
| CitcomS | Zhong 2008 | 50.21 | 0.5039 | 2.7382 | 2.7431 | -0.18 |



**Table 8.** Results for Case C1 on all meshes tested. Column labeled '% diff' represents the percent difference between top and bottom Nusselt numbers for each case. A Richardson extrapolation was applied to the different data sets to estimate the values of a theoretical mesh of infinite refinement.

| Code | Case | $\langle V_{\mathrm{RMS}} \rangle$ | $\langle T \rangle$ | $\mathrm{Nu}_t$ | $\mathrm{Nu}_b$ | % diff |
|---|---|---|---|---|---|---|
| ASPECT | 8-radial-cells | 157.103 | 0.169709 | 6.78433 | 7.15056 | -5.12 |
| | 16-radial-cells | 154.224 | 0.170962 | 7.59449 | 7.76123 | -2.15 |
| | 32-radial-cells | 154.449 | 0.171169 | 7.82858 | 7.80709 | 0.28 |
| | 64-radial-cells | 154.500 | 0.171163 | 7.82315 | 7.80388 | 0.25 |
| | Extrapolated | 154.515 | 0.171155 | 7.81417 | 7.80132 | 0.16 |
| ASPECT - EV | 8-radial-cells | 162.383 | 0.204938 | 7.85773 | 7.62804 | 3.01 |
| | 16-radial-cells | 154.445 | 0.171944 | 7.77513 | 7.77427 | 0.01 |
| | 32-radial-cells | 154.501 | 0.171145 | 7.81665 | 7.80372 | 0.17 |
| | 64-radial-cells | 154.505 | 0.171158 | 7.81749 | 7.80390 | 0.17 |
| | Extrapolated | 154.502 | 0.171176 | 7.81662 | 7.80317 | 0.17 |
| CitcomS | 24-radial-elements | 154.293 | 0.175811 | 7.05870 | 7.50938 | -6.00 |
| | 32-radial-elements | 153.046 | 0.177354 | 7.37245 | 7.56101 | -2.49 |
| | 48-radial-elements | 154.474 | 0.174750 | 7.75708 | 7.75232 | 0.06 |
| | 64-radial-elements | 154.640 | 0.172404 | 7.83635 | 7.78476 | 0.66 |
| | 96-radial-elements | 154.162 | 0.171403 | 7.83931 | 7.79612 | 0.55 |
| | Extrapolated | 153.942 | 0.171024 | 7.83836 | 7.80032 | 0.49 |
| CitcomS | Zhong 2008 | 154.8 | 0.1728 | 7.8495 | 7.7701 | 1.02 |



**Table 9.** Results for Case C2 on all meshes tested. Column labeled '% diff' represents the percent difference between top and bottom Nusselt numbers for each case. A Richardson extrapolation was applied to the different data sets to estimate the values of a theoretical mesh of infinite refinement.

| Code | Case | $\langle V_{\mathrm{RMS}}\rangle$ | $\langle T \rangle$ | $\mathrm{Nu}_t$ | $\mathrm{Nu}_b$ | % diff |
|---|---|---|---|---|---|---|
| ASPECT | 8-radial-cells | 122.061 | 0.186610 | 6.07324 | 6.74851 | -10.0 |
| | 16-radial-cells | 121.156 | 0.187833 | 6.83965 | 7.04230 | -2.88 |
| | 32-radial-cells | 121.379 | 0.188050 | 7.05674 | 7.06200 | -0.07 |
| | Extrapolated | 121.493 | 0.188114 | 7.13137 | 7.06379 | 0.96 |
| CitcomS | 24-radial-elements | 123.589 | 0.196793 | 6.70855 | 6.93783 | -3.30 |
| | 32-radial-elements | 123.773 | 0.198792 | 7.01282 | 7.05973 | -0.66 |
| | 48-radial-elements | 128.398 | 0.225521 | 7.46827 | 7.37400 | 1.28 |
| | 64-radial-elements | 125.839 | 0.209374 | 7.30565 | 7.23952 | 0.91 |
| | 96-radial-elements | 121.577 | 0.199198 | 7.00253 | 6.96235 | 0.58 |
| | Extrapolated | 119.739 | 0.195129 | 6.87126 | 6.84183 | 0.43 |
| CitcomS | Zhong 2008 | 122.1 | 0.1908 | 7.0968 | 7.0505 | 0.66 |





**Table 10.** Results for Case C3 on all meshes tested. Column labeled '% diff' represents the percent difference between top and bottom Nusselt numbers for each case. A Richardson extrapolation was applied to the different data sets to estimate the values of a theoretical mesh of infinite refinement.

| Code | Case | $\langle V_{\mathrm{RMS}} \rangle$ | $\langle T \rangle$ | $\mathrm{Nu}_t$ | $\mathrm{Nu}_b$ | % diff |
|---|---|---|---|---|---|---|
| ASPECT | 8-radial-cells | 108.445 | 0.196809 | 5.76666 | 6.48003 | -11.0 |
| | 16-radial-cells | 108.037 | 0.197600 | 6.50256 | 6.70382 | -3.00 |
| | 32-radial-cells | 108.261 | 0.197918 | 6.71399 | 6.72022 | -0.09 |
| | Extrapolated | 108.365 | 0.198035 | 6.78691 | 6.72217 | 0.96 |
| CitcomS | 24-radial-elements | 110.518 | 0.206552 | 6.62035 | 6.46213 | 2.45 |
| | 32-radial-elements | 110.441 | 0.210318 | 6.69305 | 6.69664 | -0.05 |
| | 48-radial-elements | 108.871 | 0.202368 | 6.69639 | 6.74985 | -0.79 |
| | 64-radial-elements | 108.561 | 0.200052 | 6.71333 | 6.74349 | -0.45 |
| | 96-radial-elements | 107.805 | 0.198267 | 6.71572 | 6.71589 | -0.003 |
| | Extrapolated | 107.473 | 0.197531 | 6.71627 | 6.70368 | 0.19 |
| CitcomS | Zhong 2008 | 109.4 | 0.2011 | 6.7572 | 6.7182 | 0.58 |