# Peer review of "A Comparison of 3-D Spherical Shell Thermal Convection results at Low to Moderate Rayleigh Number using ASPECT (version 2.2.0) and CitcomS (version 3.3.1)"

_Geoscientific Model Development, 2022_

## Author Response (AR1)

To whom it may concern,

The authors would like to extend thanks to everyone who commented. The following are the points discussed by referees, author responses, and how they were addressed.

The first referee is Christian Hüttig:

- Comment 1: "Although equation 5 states this, it is usually important to point out that the Rayleigh number is defined with the viscosity at T=0.5"
- Response 1: "Your remarks will be added to the manuscript." This was a minor point, no direct response was given.
- Changes 1: This information was added just after the definition of the Rayleigh number (equation 4) in the Method section.

- Comment 2: "Please cite  http://dx.doi.org/10.1016/j.pepi.2013.04.002 as we did a similar study with the CITCOM benchmark."
- Response 2: "Your remarks will be added to the manuscript." This was a minor point, no direct response was given.
- Changes 2: http://dx.doi.org/10.1016/j.pepi.2013.04.002 was added to the bibliography and cited in the Introduction.

- Comment 3: "The usability of ASPECT for 3D spherical shells seems questionable as you write in 3.1 that the case A8/9 (viscosity contrasts of 10^6 and 10^7) would have required excessive computing resources. Given that this is a benchmark from 2008 and computers got a bit faster nowadays, it seems there is an issue. If this is not the main use-case of the code this is fine, but people need to be aware of that in order to not waste computing resources and help the environment."
- Response 3: "In regards to your comment, there are several ways to improve performance for larger 3-D problems. ASPECT can use adaptive refinement to resolve features dynamically or statically which can reduce computing resources. ASPECT also has a newer geometric multigrid solver (see https://doi.org/10.1002/nla.2375 ) that can speed up computation by a factor of 3. These points will also be added to the manuscript as a short discussion."
- Changes 3: These same points were added into the Conclusions section.

The second referee is anonymous:

- Comment 1: "Some time is spent comparing and discussing different stabilization techniques but did the authors find that stabilization was actually necessary at these low Rayleigh numbers?"
- Response 1: "For your question on stabilization, no tests were performed without stabilization. These cases were run with as many default parameters as possible, so default stabilization was used for both codes."
- Changes 1: No changes were made in response to this comment.

- Comment 2: "Since one of the response to a previous review mentioned that it might be, would it be possible to include a demonstration of ASPECT at more extreme viscosity

contrasts/Rayleigh number even if a comparison to CitcomS isn't possible? This could prove useful for future code comparisons using this paper as a benchmark."

- Response 2: "For higher viscosity cases, these would need significant computational resources, which is not feasible at this time. This could be performed as a future work."
- Changes 2: No changes were made in response to this comment.

- Comment 3: "Rather than just comparing the '%diff' in top and bottom Nusselt numbers in the results tables could some quantitative comparisons be made between the codes to clearly demonstrate their similarity."
- Response 3: "For further comparison between the codes, the reported parameters were chosen to best match the Zhong et al., 2008 benchmark paper. Radial plots, isotherms and Degrees of Freedom for the various mesh refinements used are all provided as comparisons."
- Changes 3: No changes were made in response to this comment.

- Comment 4: "line 27: presumably the length-scale is D not D^3"
- Response 4: "Your minor points have also been added, thank you so much for the variety of typos pointed out." This was a minor point, no direct response was given.
- Changes 4: At the end of the Introduction, $D^3$ was changed to $D$.

- Comment 5: "line 9: the Rayleigh number alone doesn't describe the problem as it also depends on the choice of boundary conditions, which are given later but perhaps this sentence could be rephrased to "The Rayleigh number (plus apropriate boundary conditions) can describe this problem..."?"
- Response 5: "Your minor points have also been added, thank you so much for the variety of typos pointed out." This was a minor point, no direct response was given.
- Changes 5: This sentence was changed as suggested and can be found just before the definition of the Rayleigh number (equation 4) in the Method section.

- Comment 6: "line 21: only equations (1-3) are solved, not (1-4), which includes the definition of Ra"
- Response 6: "Your minor points have also been added, thank you so much for the variety of typos pointed out." This was a minor point, no direct response was given.
- Changes 6: This line in the Method section was changed to say equations 1-3 as suggested.

- Comment 7: "equation 5/table 2: it would be a convenience to those trying to reproduce this to provide the values of E used in table 2 rather than just the viscosity differences"
- Response 7: "Your minor points have also been added, thank you so much for the variety of typos pointed out." This was a minor point, no direct response was given.
- Changes 7: A more explicit definition for total viscosity variation involving E was added in the paragraph defining the viscosity in the Method section.

- Comment 8: "equation 10: assuming Omega is the volume of the domain (is this defined somewhere?) it seems unusual that the definition of RMS velocity uses the square root of Omega, is this a typo?"

- Response 8: "For the question on equation 10, this is not a typo, and the volume is defined in equation 8."
- Changes 8: No changes were made in response to this comment.

- Comment 9: "line 23: "This allowed isolate" -> "This allowed us to isolate"?  This paragraph feels a bit repetetive of the previous paragraph, which also discusses the use of uniformly-spaced meshes in the radial direction."
- Response 9: "Your minor points have also been added, thank you so much for the variety of typos pointed out." This was a minor point, no direct response was given.
- Changes 9: After the paragraph defining the Nusselt numbers, volume, average temperature, and VRMS in the Method section (equations 6-10), the next two paragraphs were merged and rewritten to address this comment.

- Comment 10: "line 27: the doi given for Bangerth et al. (2020b) only appears to go up to Figure 123, could you please check this reference to Figure 130?"
- Response 10: "Your minor points have also been added, thank you so much for the variety of typos pointed out." This was a minor point, no direct response was given.
- Changes 10: A second reference to the most current ASPECT User Manual was added, and this sentence reworded to refer to the section name in the manual, rather than the figure number as those change more frequently. This can be found in the first paragraph of the Results section.

- Comment 11: "line 23: please clarify the phrase "ASPECT is very likely converging with higher mesh resolution for this parameter as well""
- Response 11: "Your minor points have also been added, thank you so much for the variety of typos pointed out." This was a minor point, no direct response was given.
- Changes 11: This sentence at the end of the 3-D Results for the Stagnant-Lid Case: A7 was rewritten for clarity.

- Comment 12: "table 1: units for gravitational acceleration are incorrect"
- Response 12: "Your minor points have also been added, thank you so much for the variety of typos pointed out." This was a minor point, no direct response was given.
- Changes 12: The units for gravity given in Table 1 were corrected to $m/s^2$.

- Comment 13: "tables 5-10: incorrect quotation mark used around '% diff' in all captions"
- Response 13: "Your minor points have also been added, thank you so much for the variety of typos pointed out." This was a minor point, no direct response was given.
- Changes 13: The captions for Tables 5-10 all had this piece of text fixed.

Again, many thanks to our reviewers. Please let me know if there's anything else that can be made clear,

Grant Euen